# Performance Characteristics of the Ultrasound Strategy during Incidence Screening in the UK Collaborative Trial of Ovarian Cancer Screening (UKCTOCS)

**DOI:** 10.3390/cancers13040858

**Published:** 2021-02-18

**Authors:** Jatinderpal Kalsi, Aleksandra Gentry-Maharaj, Andy Ryan, Naveena Singh, Matthew Burnell, Susan Massingham, Sophia Apostolidou, Aarti Sharma, Karin Williamson, Mourad Seif, Tim Mould, Robert Woolas, Stephen Dobbs, Simon Leeson, Lesley Fallowfield, Steven J. Skates, Mahesh Parmar, Stuart Campbell, Ian Jacobs, Alistair McGuire, Usha Menon

**Affiliations:** 1Department of Women’s Cancer, Institute for Women’s Health, University College London, London WC1E 6HU, UK; j.k.kalsi@ucl.ac.uk (J.K.); i.jacobs@unsw.edu.au (I.J.); 2MRC Clinical Trials Unit at UCL, Institute of Clinical Trials & Methodology, London WC1V 6LJ, UK; a.gentry-maharaj@ucl.ac.uk (A.G.-M.); a.ryan@ucl.ac.uk (A.R.); m.burnell@ucl.ac.uk (M.B.); s.massingham@ucl.ac.uk (S.M.); s.apostolidou@ucl.ac.uk (S.A.); m.parmar@ucl.ac.uk (M.P.); 3Department of Pathology, Barts and the London, London E1 2ES, UK; Naveena.Singh@bartshealth.nhs.uk; 4Department of Obstetrics and Gynaecology, University Hospital of Wales, Cardiff CF14 4XW, UK; Aarti.Sharma@wales.nhs.uk; 5Department of Gynaecological Oncology, Nottingham City Hospital, Nottingham NG5 1PB, UK; Karin.Williamson@nuh.nhs.uk; 6Division of Gynaecology and of Cancer Services, St. Mary’s Hospital and University of Manchester, Manchester M13 9WL, UK; Mourad.Seif@mft.nhs.uk; 7Department of Gynaecological Oncology, University College Hospital, London NW1 2BU, UK; tim.mould@uclh.nhs.uk; 8Department of Gynaecological Oncology, Queen Alexandra Hospital, Portsmouth PO6 3LY, UK; robert.woolas2@porthosp.nhs.uk; 9Department of Gynaecological Oncology, Belfast City Hospital, Belfast BT9 7AB, UK; Stephen.Dobbs@belfasttrust.hscni.net; 10Department of Obstetrics and Gynaecology, Ysbyty Gwynedd, Bangor, Gwynedd LL57 2PW, UK; simon.leeson@wales.nhs.uk; 11Cancer Research UK Sussex Psychosocial Oncology Group at Brighton & Sussex Medical School, University of Sussex, Falmer BN1 9PX, UK; L.J.Fallowfield@sussex.ac.uk; 12Massachusetts General Hospital, Harvard Medical School, Boston, MA 02115, USA; sskates@gmail.com; 13Create Fertility Clinic, London EC2V 6ET, UK; profscampbell@hotmail.com; 14Department of Women’s Health, University of New South Wales, Australia, Sydney 2052, Australia; 15London School of Economics and Political Science, London WC2A 2AE, UK; A.J.Mcguire@lse.ac.uk

**Keywords:** ovarian cancer, screening, ultrasound, TVS, early detection, trial, randomised controlled trial, UKCTOCS

## Abstract

**Simple Summary:**

The United Kingdom Collaborative Trial of Ovarian Cancer Screening was undertaken to assess whether screening postmenopausal women from the general population might result in detection of ovarian/tubal cancers at an earlier stage and thus save lives. One of the screening strategies tested was a yearly transvaginal ultrasound scan of the ovaries (USS). Following the initial screen, 44,799 of the 50,639 women in the USS group went on to have a further 280,534 annual scans during April 2002–December 2011. Abnormalities leading to surgery were detected in 960 women of whom 113 (80 invasive epithelial) had ovarian/tubal cancer. Ovarian/tubal cancer was missed in 52 (50 invasive epithelial) women. Of the screen-detected cancers, 37.5% and missed cancers 6% were early stage(I/II). The number (detection rate 61.5%; 80/130) and advanced stage of the missed invasive cancers suggests that a yearly ultrasound scan may not be suitable for screening average risk women for ovarian cancer.

**Abstract:**

Randomised controlled trials of ovarian cancer (OC) screening have not yet demonstrated an impact on disease mortality. Meanwhile, the screening data from clinical trials represents a rich resource to understand the performance of modalities used. We report here on incidence screening in the ultrasound arm of UKCTOCS. 44,799 of the 50,639 women who were randomised to annual screening with transvaginal ultrasound attended annual incidence screening between 28 April 2002 and 31 December 2011. Transvaginal ultrasound was used both as the first and the second line test. Participants were followed up through electronic health record linkage and postal questionnaires. Out of 280,534 annual incidence screens, 960 women underwent screen-positive surgery. 113 had ovarian/tubal cancer (80 invasive epithelial). Of the screen-detected invasive epithelial cancers, 37.5% (95% CI: 26.9–49.0) were Stage I/II. An additional 52 (50 invasive epithelial) were diagnosed within one year of their last screen. Of the 50 interval epithelial cancers, 6.0% (95% CI: 1.3–16.5) were Stage I/II. For detection of all ovarian/tubal cancers diagnosed within one year of screen, the sensitivity, specificity, and positive predictive values were 68.5% (95% CI: 60.8–75.5), 99.7% (95% CI: 99.7–99.7), and 11.8% (95% CI: 9.8–14) respectively. When the analysis was restricted to invasive epithelial cancers, sensitivity, specificity and positive predictive values were 61.5% (95% CI: 52.6–69.9); 99.7% (95% CI: 99.7–99.7) and 8.3% (95% CI: 6.7–10.3), with 12 surgeries per screen positive. The low sensitivity coupled with the advanced stage of interval cancers suggests that ultrasound scanning as the first line test might not be suitable for population screening for ovarian cancer. Trial registration: ISRCTN22488978. Registered on 6 April 2000.

## 1. Introduction

Transvaginal ultrasonography (TVS) is considered the best modality for pelvic imaging, and is used routinely in the clinic for investigating women with suspected ovarian cancer. Based on its ability to assess ovarian volume and morphology, it has been used in large randomised trials of ovarian cancer screening as the primary screen. In the ovarian arm of the Prostate Lung Colorectal and Ovarian (PLCO) Cancer Screening trial [1], it was used in combination with the serum biomarker CA125 while in the ultrasound arm (USS) of the United Kingdom Collaborative Trial of Ovarian Cancer Screening (UKCTOCS), it was used as the sole primary screening test [2]. In both trials, there was no difference in the proportion of women detected with Stage I/II disease or deaths due to ovarian/tubal/peritoneal cancer between the ultrasound arm and the no screening (control) arm [2].

The data collected during these trials provides a rich resource to understand the performance characteristics of TVS in the setting of multicentre, general population screening. We have previously reported on the results of the initial (prevalence) USS screen [3]. We now report on the performance characteristics of USS screening in UKCTOCS during the 10 years of incidence screening. 

## 2. Results

Following the initial (prevalence) screen, of the 50,639 women randomised to the USS arm 49,610 were eligible for incidence screening. Of them, 1029 were ineligible as both ovaries had been removed (896), death (131), moved away (2). Overall, 44,799 (88.5%) of those randomized to the USS arm underwent incidence screening (Figure 1). 

In total the women underwent 280,534 annual incidence screens between 28 April 2002 and 31 December 2011. Of these screens, 257,337 (91.8%) were TVS, 20,707 (7.4%) transabdominal, 2309 (0.8%) both and for nine data on mode were missing. Individual women attended between 1 and 10 incidence screens with the median number per woman being 7 (IQR 5–8). The baseline characteristics of these women have been previously reported [2,3]. Median age of the women at the last annual incidence screen was 67 (IQR 62.6–72.0) years. 

Overall, 99.4% (278,851/280,534) of the screens resulted in women being returned to annual screening. Two percent (5497/280,534) of screens involving 4256 (9.5%; 4256/44,799) women resulted in referral for clinical evaluation. Of these women 960 (0.34% of screens; 960/280,534) were screen positive and had surgery (Figure 2 and Table 1). This figure includes one woman with a simple ovarian cyst who underwent surgery against protocol recommendation. 

Of the 960 surgical procedures, 69% (662/960) were laparoscopic or vaginal. 113 (11.8%) women were diagnosed with ovarian/tubal cancers (Table 2). This included 80 (70.8%) invasive epithelial ovarian or tubal (iEOC), 29 (25.7%) borderline (low malignant potential) epithelial ovarian, and 4 (3.5%) non-epithelial ovarian cancers. 

Of the 29 borderline epithelial ovarian cancers, 28 (96.5%) were Stage I/II as were 3 of 4 (75%) non-epithelial ovarian cancers. Of the screen detected iEOC, 37.5% (30/80) were Stage I/II (Table 3). Of the iEOC 80% (64/80) were Type II and 18.8% (15/80) were Type I. Majority (86.7%; 13/15) of Type I were Stage I/II. Of Type II, only 26.6% (17/64) were Stage I/II. The median time from Level 1 annual screen to surgery for screen detected iEOC was 12.6 weeks (IQR 8.7 to 20.5). 

Of the 960 women who had screen positive surgery, 831 had benign pathology or normal adnexa (Table 2). In this subgroup, 35 (4.2%) women had a major complication (with significant sequelae) (Appendix A). 

Median follow up from the end of incidence screening to cancer registration update in 2015 (25 March 2015 England and Wales, 15 April 2015 Northern Ireland) was 3.9 (IQR 3.6–5.0) years. Only 5 of 44,799 (0.01%) women had follow-up of less than 2 years after their last screen. An additional 52 women were diagnosed with ovarian/tubal cancer (screen negative/interval cancer) within 1 year of the last incidence screen scan (Table 2). This included 2 borderline and 50 iEOC. Of the latter, 6% (3/50) were diagnosed at Stage I/II (Table 3). 

The sensitivity, specificity, and positive predictive values (PPV) were 68.5% (95% CI: 60.8–75.5), 99.7% (95% CI: 99.7–99.7), and 11.8% (95% CI: 9.814) respectively for all ovarian and tubal cancers with 8.5 operations per case detected during incidence screening. When the analysis was restricted to iEOC, sensitivity, specificity and positive predictive values were 61.5% (95% CI: 52.6–69.9); 99.7% (95% CI: 99.7–99.7); and 8.3% (95% CI: 6.7–10.3) with 12 surgeries per screen positive (Table 4). 

Combining incidence and prevalence screening [3] of UKCTOCS, the sensitivity, specificity and positive predictive values were 72.3% (95% CI: 65.9–78.0), 99.5% (99.5–99.5), and 9.1% (95% CI: 7.8–10.5) for all ovarian and tubal cancers with 11.0 operations per case detected. When the analysis was restricted to iEOC, sensitivity, specificity, and positive predictive values were 63.3% (55.4, 70.6), 99.5% (95% CI: 99.5–99.5), and 5.8% (4.78–7) with 17.2 surgeries per screen positive. 

## 3. Discussion

### 3.1. Principal Findings

The performance characteristics of ultrasound screening in the largest ovarian cancer screening trial suggests that USS may not be suitable as a first line test for population screening. While the PPV was significantly higher (11.8% vs. 5.3%; *p* < 0.0001) with fewer operations (8.5 vs. 18.8; *p* < 0.0001) required to detect an ovarian/tubal cancer during incidence screening compared to the prevalence [3], the sensitivity was lower (68.5% versus 84.9%; *p* = 0.02). For invasive epithelial cancers, while over one-third (38%) of the screen detected invasive cancers were early stage, the majority (94.0%) of the interval cancers were advanced (Stage III/IV). The latter, coupled with the low sensitivity (61.5%) resulted in no overall difference (24% USS versus 26% Control; *p* = 0.57) in low volume (Stage I, II, IIIa) invasive epithelial disease between USS and control arm on the previously reported intention to treat analysis [1,2]. 

### 3.2. Results in Context 

While TVS is integral to all ovarian cancer screening strategies to date, its use as the primary screening test, as described here, has only been assessed in one other study, the University of Kentucky Ovarian Cancer Screening Trial (UKOCST). The latter study involved a slightly higher risk population with just under one fourth having a family history of ovarian and over 40% of breast cancer. It is a single arm single-centre prospective study and involved 46,101 women who underwent a mean of 6.5 annual screens [4]. Overall sensitivity for detecting ovarian cancers (85.5% vs. 72%) was higher than in the USS arm of our trial. TVS has a significant subjective component that is likely to be the key contributor to the differences noted. UKOCST involves a single centre, with all scans performed by a small group of highly experienced ultrasonologists. UKCTOCS involved over 200 Level I ultrasonologists [5] (certified sonographers or doctors with experience in gynaecological scanning in the National Health Service) across 13 centres undertaking ~45,000 scans every year. The latter is more akin to a general population screening programme which would require annual scans for millions of women.

The sensitivity in our USS arm was significantly higher than that sensitivity of TVS alone (44.6%; 33/74) noted during four rounds of screening in the PLCO trial [6]. In the latter trial, the overall sensitivity was higher as the annual screen involved CA125 in addition to a scan, with abnormalities in both tests triggering additional investigations (combined strategy). 

In comparison to a CA125-based strategy, PPV of ultrasound screening is low. The number of operations per ovarian cancer decreased from 18.8 during prevalence screening in our USS arm to 8.5 during incidence screening. This latter is similar to the 7.4 operations per case reported in the Kentucky study [4]. It is not possible to calculate a comparable estimate in the PLCO trial as a combined strategy was used. 

In our trial, 10,000 complex adnexal masses were detected during the annual incidence screen. Through a process of repeat scanning for persistence of lesion and evaluation of ultrasound features by Level 2 expert sonologists, we were able to restrict surgery to just below 1000 of these women. Both the Kentucky and International Ovarian Tumour Analysis (IOTA) groups have over the years developed increasingly sophisticated rules/scoring systems to improve risk stratification of these adnexal masses and encourage conservative management. In the most recent international IOTA5 study of women with adnexal masses, they were able to avoid surgery in one-third on the basis of low risk ultrasound features [7]. 

A key requirement to impact on the high ovarian cancer mortality is detection of invasive epithelial ovarian/tubal cancer at a sufficiently early stage. A similar proportion of screen detected ovarian cancers were invasive epithelial both in our analysis (71%: 80/113) and in the Kentucky study (75.5%; 71/94). However, only 37.5% (95% CI: 26.9, 49.0) of screen detected invasive epithelial cancers were early stage (I and II) in our trial compared to 51% (45/71; 95% CI: 51.1, 74.5) in the latest report of the Kentucky study [4]. In the latter, this together with increased sensitivity is likely responsible for the significantly higher 5-year disease-specific survival of women with ovarian (including interval) cancers in the screening group (79 ± 4%) compared to unscreened women with clinically detected epithelial ovarian cancer treated at the same centre during the same time period (45 ± 2%).

In comparison to a CA125 based approach [3], an ultrasound-based strategy detects a larger proportion of borderline ovarian cancers. This was similar in the Kentucky study (15.5%; 17/124, 95% CI: 9.3,23.6) and during incidence screening in UKCTOCS (18.8%; 29/165,95% CI: 13.1, 25.6). In our prevalence screen, it was higher (37.7%, 20/53, 95% CI: 24.7, 52.1). The lower incidence with time is likely due to increasing conservative management of less complex asymptomatic adnexal masses. 

### 3.3. Clinical and Research Implications

The performance characteristics suggest that ultrasound as a first line test is not suitable for population ovarian cancer screening. The subjective nature of TVS, the challenges in identifying normal postmenopausal ovaries [8] that diminish in size with age and the low disease prevalence (1 in 2500) means that detection of disease early requires significant expertise coupled with constant attention to detail. In the course of the trial, we developed an accreditation programme for scanning postmenopausal ovaries [5]. However, our performance characteristics suggest that we were not able to replicate in the Level 1 ultrasonographers, the expertise available at a specialist centre such as Kentucky. The IOTA group have shown in multicentre studies that the performance of ultrasound prediction models/rules can be maintained in sonographers with varying levels of experience [9]. However, this is in the context of evaluation of adnexal masses, which is equivalent to a Level 2 rather than Level 1 screen during population screening. First line TVS screening of the population is always going to be a challenge given the size of the workforce required. The ideal is a less subjective, automated, and more reproducible test. In cervical screening, this has translated to HPV DNA testing increasingly replacing the older resource intensive and skill dependent cytology in many population-screening programmes.

Incidental adnexal findings are on the rise given the widespread use of ultrasound. The unnecessary surgery rates seen in our and the other ultrasound screening trials are relevant to the clinical management of these asymptomatic masses. Our findings suggest that many with low-risk features can be managed conservatively [10]. 

### 3.4. Strengths and Limitations

The key strengths of our study are the scale of the trial, high compliance with screening, the multicentre setting and detailed screening protocols and automated management algorithms, implemented by a dedicated central team. Completeness of data on screen-negative cancers was ensured by flagging of the trial cohort through cancer, death, and hospital administrative registries as well as postal follow-up of all women. All potential ovarian cancer cases were reviewed by an independent, blinded outcomes review committee. 

A key limitation relates to use of self-reported visualisation of postmenopausal ovaries as a quality assurance measure during the trial. A retrospective audit of random, grey scale TVS images showed only moderate agreement for visualisation of normal ovaries between experts and sonographers and between expert reviewers alone [8]. This was despite a robust accreditation programme established within the trial for visualisation of postmenopausal ovaries. This again highlights the subjectivity of ultrasound scanning, use of video recordings of the ultrasound examination would probably have been a better-quality assurance measure. During the 14 years of trial, there have been significant advances in our understanding of the origin and heterogeneity of ovarian cancer. Our scanning protocol focused on evaluation of the ovary. However, we now know that at least half of high-grade serous cancers arise in the fallopian tube [11] making tubal evaluation critical. The Kentucky group has recently described and assessed such a protocol in older normal women and reported a 77% visualisation rate [12]. Furthermore, in the last decade, there has been significant improvement in the resolution of ultrasound machines and their ability to detect subtle changes as a result of advances in ultrasound transducer technology and electronics. 

## 4. Materials and Methods 

### 4.1. Ethical Approval

The trial (ISRCTN22488978, ClinicalTrials.gov NCT00058032) was approved by the UK North West Multicentre Research Ethics Committees (North West MREC 00/8/34) with site specific approval from the local regional ethics committees and the Caldicott guardians (data controllers) of the primary care trusts. All participants provided written consent.

### 4.2. Subjects and Screening Strategy

The trial design has been described previously [2,3,13]. Briefly, 202,638 postmenopausal women aged 50 to 74, from the general population were recruited through 13 regional trial centres located in NHS Trusts in England, Wales and Northern Ireland, between April 2001 and October 2005. Overall, 1.6% of women had a maternal history of ovarian cancer and 6.3% a maternal history of breast cancer [3]. Women at increased risk of familial ovarian cancer were excluded from the study. The participants were randomised 1:1:2 to annual screening (until 31 December 2011) with serum CA125 (MMS: 50, 640) or TVS (USS: 50, 639) or no screening (control C: 101, 359). The full trial protocol is accessible at http://ukctocs.mrcctu.ucl.ac.uk/media/1066/ukctocs-protocol_v90_19feb2020.pdf (accessed on 4 February 2021). In the USS arm, 48,230 women underwent an initial (prevalence) screen [3].

Scans were performed by trial sonographers, the majority of whom worked in the NHS providing gynaecological scanning. All trial sonographers underwent additional training for assessment of postmenopausal ovaries and from 2008, formal accreditation [5]. Annual (Level 1) scans were performed by Type 1 (certified sonographers, trained midwives, or doctors with experience in gynaecological scanning) or Type 2 (experienced gynaecologists/radiologists, or senior sonographers, usually superintendent grade with particular expertise in gynaecological scanning) ultrasonographers. Repeat scans on detection of an abnormality (Level 2 scans) were only undertaken by Type 2 sonographers. Most scans during 2002–2008 were done on a dedicated Kretz SA9900 ultrasound machine (Medison, Seoul, Korea) and from 2008–2011 on Acquvix (Medison, Seoul, Korea).

At the annual transvaginal scan (Level 1), ovarian morphology and dimensions were assessed, and ovarian volume calculated. Ovarian morphology was classified as normal, simple cyst (single, thin walled, anechoic cyst with no septa or papillary projections) or complex (ovary had any non-uniform ovarian echogenicity excluding single simple or inclusion cyst). The number and size of cysts, wall regularity, presence and thickness of septae, size of papillations, and echogenicity of the fluid contents were recorded. The cysts were initially classified using the Kentucky screening trial morphology index [14] and from 2003, the International Ovarian Tumour Analysis (IOTA) classification [15]. Where an ovary was not visualised, the sonographer documented ‘good view’ if 3–5 cms of iliac vessels with well-defined walls and a clear anechoic centre was seen or ‘poor view’ and stated the reason such as bowel, fibroids, pelvic varicosities, or other. Ascites was defined as a vertical pool of fluid measuring >10 mm in the Pouch of Douglas. 

Ultrasound scans were classified based on the morphology of the adnexa and visualisation of the surrounding tissue as follows: (a) normal—where both ovaries had normal morphology or simple cysts were <60 cm^3^, or were not visualised but a good view of the iliac vessels was obtained; (b) unsatisfactory—where one or both ovaries were not visualised due to a poor view); (c) abnormal—where one or both ovaries had complex morphology or simple cysts were >60 cm^3^, or ascites was present. Based on these results the women were returned to annual screening (normal scan), repeat Level 1 scan (unsatisfactory scan) or Level 2 scan (abnormal scan). In women where adnexal masses had been previously managed conservatively and remained unchanged in morphology or volume (complex unchanged) on repeat annual screens, there was the option for clinical review of results and return to annual screening without undergoing Level 2. Women with an abnormal Level 2 scan were referred for clinical assessment.

This was undertaken at the regional centre by a designated trial clinician and included clinical evaluation and investigations as appropriate. Latter included serum CA125, repeat transvaginal scans and Doppler studies, CT/MRI of the abdomen and pelvis, and occasionally assessment of other tumour markers. A decision was made either to offer surgery or manage conservatively, taking into account the views of the woman, any significant comorbidity, morphological features of the ultrasound-detected lesion, previous hysterectomy, or major pelvic surgery that could contribute to false-positive ultrasound findings. The surgery in most cases involved removal of both ovaries and fallopian tubes using a laparoscopic approach where possible. If pelvic adhesions increased the risk of complications, the clinician could opt to remove only the ‘abnormal’ ovary. Hysterectomy was only undertaken where there was clear clinical indication. Women found to have ovarian or tubal cancer at a primary laparoscopic procedure underwent a subsequent staging procedure. Where there was high suspicion of ovarian cancer, laparotomy was undertaken. For those managed conservatively, the follow up plan usually involved a TVS and serum CA125 at 3 months with a possible repeat at 6 months, and return to annual screening if the findings were unchanged (unchanged complex). 

### 4.3. Follow-Up

Follow up involved electronic health record linkage for cancer and death registration and hospital admissions using the NHS number through the appropriate national agencies. Cancer registrations received until 25 March 2015 (England and Wales) and 15 April 2015 (Northern Ireland) were used for this analysis. In addition, women were sent postal questionnaires, 3–5 years post randomisation, and again in April 2014 after the end of screening [2]. 

### 4.4. Confirmation of Diagnosis

Copies of medical notes were retrieved for all women who had surgery as a consequence of a positive screening test as previously described [2]. Where cancer was diagnosed, additional information e.g., multidisciplinary team meeting notes, discharge summaries, and other relevant correspondence was also collated. The above were also obtained for all women where a notification was received either through linked electronic health records, follow-up questionnaire, or personal communication of a possible ovarian/tubal/peritoneal cancer. The case notes of all of these individuals were reviewed by an Outcomes Review Committee blinded to the randomisation group. They confirmed primary site, stage, morphology, and—where possible—classified invasive epithelial cancer into Type I (low-grade serous, low-grade endometrioid, mucinous, and clear cell cancers) or Type II (high-grade serous, high-grade endometrioid, carcinosarcomas and undifferentiated carcinoma) cancers [16]. Primary site was originally classified according to WHO 2003 [17] and more recently revised using WHO 2014 classification [18]. As a result, cancers initially classified as peritoneal have been reclassified for this analysis as ovarian/tubal. Stage for all cases included in this analysis have been re-reviewed by the Outcomes Committee and assigned as per FIGO 2014 criteria [19].

### 4.5. Analysis

This analysis is limited to annual screens that followed the initial (prevalence screen). An annual screen as previously defined is a single or series of scans culminating in surgery (screen positive) or return to annual screening (screen negative). For this analysis, women were censored at one year following the last scan performed as part of their last screening episode on the trial. The primary outcome measure was ovarian or fallopian tube cancer as per WHO 2014 classification [18] diagnosed within 12 months of the last scan. Sensitivity (proportion of ovarian/tubal cancers diagnosed within one year that were detected by screening), specificity (proportion of those without ovarian/tubal cancer who had a negative screen) and positive predictive value (proportion with a positive test result who actually had ovarian/tubal cancer) of incidence screening was calculated. Subgroup analysis of invasive epithelial cancers (borderline epithelial and non-epithelial ovarian cancers were excluded) was undertaken. Proportion of cancers detected in early (I/II) stage were calculated. 

## 5. Conclusions

The performance characteristics suggest that ultrasound as the first line test may not be suitable for population screening.

## Figures and Tables

**Figure 1 cancers-13-00858-f001:**
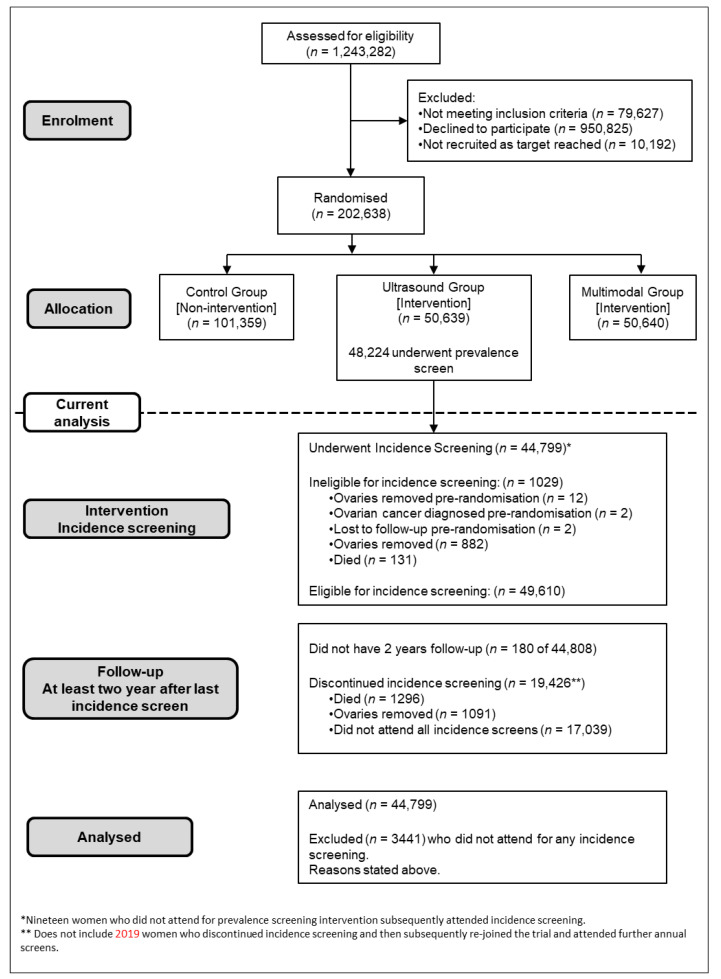
CONSORT diagram.

**Figure 2 cancers-13-00858-f002:**
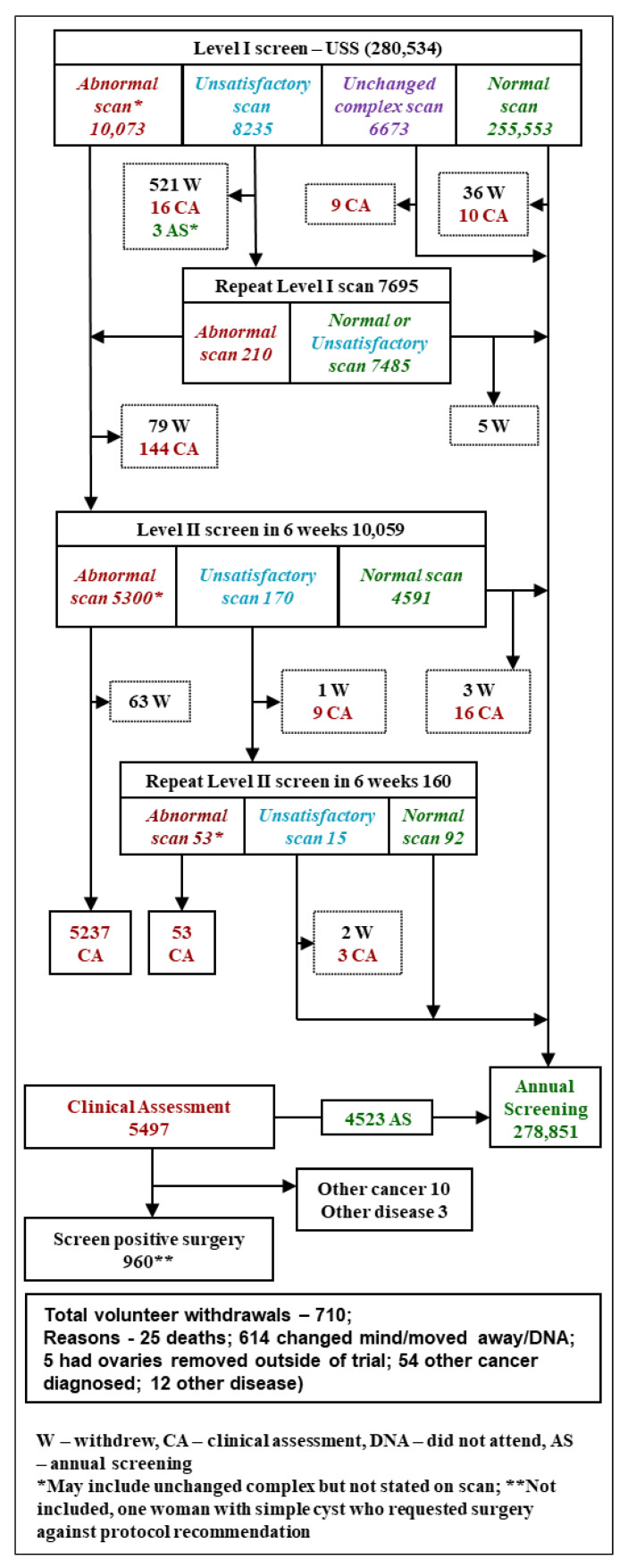
Ultrasound screening (USS) algorithm and outcome of incidence screening.

**Table 1 cancers-13-00858-t001:** Results of annual incidence screens performed in USS group.

Annual Incidence Screens		Women Years
No. of Level 1 Screens *		280,534 (100)
	Normal Scan	262,227 (93.5)
	Unsatisfactory Scan	8235 (2.9)
	Abnormal Scan	10,072 (3.6)
No. who Underwent Repeat Level 1 Screen †		7695 (2.7)
	Returned to annual screening	7485 (97.3)
	Referred for Level 2 screen	210 (2.7)
No. Who Underwent Level 2 Screen †		10,060 (3.6)
	Returned to annual screening	4591 (45.6)
	Referred for clinical assessment	5299 (52.7)
	Referred for Repeat Level 2 screen	170 (1.7)
No. Who Underwent Repeat Level 2 Screen †		160 (0.1)
	Returned to annual screening	92 (57.5)
	Referred for clinical assessment	68 (42.5)
No. Referred for Clinical Assessment †,‡		5495 (2.0)
No. Who Underwent Screen Positive Surgery †		960 (0.3)
Surgical Approach		
	Diagnostic laparoscopy §	31 (3.2)
	Operative laparoscopy	628 (65.4)
	Combined laparoscopy and laparotomy	69 (7.2)
	Laparotomy	214 (22.3)
	Vaginal hysterectomy with BSO	3 (0.3)
	Imaging guided cytology/biopsy	14 (1.5)
	Missing data	1 (0.1)

Data is number (%). * Denominators for header rows are numbers of annual screens. Denominators for subsequent rows are number who underwent specific screen. † Difference in numbers between those recommended tests and number who underwent test is due to non-compliance. ‡ 123 women were clinically assessed following a level 1 screen. § Seven women went on to have laparotomy as a second procedure.

**Table 2 cancers-13-00858-t002:** Pathologic findings in screen positive women and those with interval cancers (screen negative).

Outcome of Screen Positive Surgery	All Women
**Total ***	**960**
Normal or benign pathology	831
Laparoscopy, ovaries normal, not removed	24
Normal ovaries	91
Benign ovarian pathology	716
Non-ovarian/tubal malignant neoplasms	13
Other non-ovarian cancer involving the ovaries(secondary ovarian neoplasm)	7 **
Other non-ovarian cancer not involving the ovaries	6
**Screen Positive Women Diagnosed with Malignant Neoplasm of Ovary (ICD-C56) and Fallopian Tube (ICD-C57.0)**	
**Total**	**113**
Non-epithelial neoplasm of ovary (ICD-C56)	4
Borderline epithelial neoplasm of ovary (ICD-C56)	29
Invasive epithelial neoplasm of tubo-ovarian origin (ICD-C56/C57.0)	80
**Women with Screen Negative (Interval) Malignant Neoplasm of Ovary (ICD-C56) and Fallopian Tube (ICD-C57.0) Diagnosed within One Year of End of Screen**	
**Total**	**52**
Borderline epithelial neoplasm of ovary (ICD-C56)	2
Invasive epithelial neoplasm of tubo-ovarian origin (ICD-C56/C57.0)	50

Data are numbers. * Includes one volunteer who withdrew consent for accessing medical records and two volunteers where the ovaries were not identified due to extensive adhesions arising from a previous hysterectomies. ** Cancers of colorectal (3) breast (1), stomach (1), lymphoma (1), carcinoid small bowel (1).

**Table 3 cancers-13-00858-t003:** Stage and type of invasive epithelial ovarian and tubal cancers as per WHO 2014 classification.

Characteristics	Positive	Negative
Total	80	50
FIGO 2014 Stage		
I	18	2
II	12	1
III	45	26
IIIa	5	0
IIIb	13	3
IIIc	27	23
IV	5	21
Early (I/II) stage-%(95% CI)	37.5 (26.9, 49.0)	6.0 (1.3, 16.6)
Morphology		
**Type I iEOC (total)**	**15 (18.8%)**	**1 (2.0%)**
Low grade serous	3	0
Endometrioid (low grade)	3	0
Clear cell	6	0
Mucinous	3	1
**Type II iEOC (total)**	**64 (80.0%)**	**42 (84.0%)**
High grade serous	53	36
High grade endometrioid	4	0
Carcinoma	4	6
Carcinosarcoma	3	0
Unclassified *	1 (1.3%)	7 (14.0%)

Date are numbers unless otherwise stated. * Morphology could not be determined as only peritoneal fluid cytology was undertaken.

**Table 4 cancers-13-00858-t004:** Performance characteristics of incidence USS screening for detection of ovarian and tubal cancers (WHO 2014 classification) within one year of screen.

Characteristics	No/% (95% CI)
Number of women screen years	280,534
Number of surgeries	960
**Ovarian and Tubal Malignancies**	
Screen positives	113
Screen negatives	52
Sensitivity	68.5% (60.8, 75.5)
Specificity	99.7%(99.7, 99.7)
Positive predictive value	11.8%(9.8, 14.0)
No. of operations per screen positive	8.5
**Invasive Epithelial Ovarian and Tubal Malignancies ***	
Screen positives	80
Screen negatives	50
Sensitivity	61.5% (52.6, 69.9)
Specificity	99.7% (99.7, 99.7)
Positive predictive value	8.3% (6.7, 10.3)
No. of operations per screen positive	12.0

Data are numbers or % (95% CI) * excludes non epithelial and borderline epithelial ovarian neoplasms).

## Data Availability

The datasets used and/or analysed during the current study are available from the corresponding author on reasonable request.

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
