# Peer review of "Performance Characteristics of the Ultrasound Strategy during Incidence Screening in the UK Collaborative Trial of Ovarian Cancer Screening (UKCTOCS)"

_cancers, 2021, doi:10.3390/cancers13040858_

Round 1
Reviewer 1 Report
Well written overall, and on an important subject.
Would clarify in the abstract that 12 surgeries were necessary to detect one invasive EOC. May thus add in the abstract that for invasive EOC, after the % listed for sensitivity, specificity etc, ...”with 12 surgeries per screen positive.”
Currently, the text is oddly situated in between tables, and could benefit from rearrangement, for the reader to better follow the text.
The tables are also not very clear - e.g. in Table 2, the header “screen positive women diagnosed with malignant neoplasm of ovary and Fallopian tube” should be made bold, as it appears to be a header for the data below it. To be consistent, bolden “Total” below as well.
Author Response
Reviewer 1:
(a) Would clarify in the abstract that 12 surgeries were necessary to detect one invasive EOC. May thus add in the abstract that for invasive EOC, after the % listed for sensitivity, specificity etc, ...”with 12 surgeries per screen positive.”
We thank the Reviewer for suggesting the additional text clarifying the number of surgeries required per screen positive case. Amendment as below:
Abstract:
Transvaginal ultrasound was used both as the first and the second line test.
When the analysis was restricted to invasive epithelial cancers, sensitivity, specificity and positive predictive values were 61.5%(95%CI:52.6-69.9); 99.7%(95%CI:99.7-99.7) and 8.3%(95%CI:6.7-10.3), with 12 surgeries per screen positive.
(b) Currently, the text is oddly situated in between tables, and could benefit from rearrangement, for the reader to better follow the text.
We have moved the text so that the tables and figures immediately follow the relevant text.
(c)The tables are also not very clear - e.g. in Table 2, the header “screen positive women diagnosed with malignant neoplasm of ovary and Fallopian tube” should be made bold, as it appears to be a header for the data below it. To be consistent, bolden “Total” below as well.
The tables seem to have lost the formatting that we had applied. We have now reformatted the tables, aligning the text in the 1st column to the left. We would request the journal to keep this formatting as it makes it easier to understand the table contents.
We have made header and “Total” bold in Table 2 as requested by Reviewer 1.
Reviewer 2 Report
The manuscript by Kalsi and co-authors reports on the performance of the transvaginal ultrasound strategy, used as both first and second line test, in the incidence screening of the UK Collaborative Trial of Ovarian Cancer Screening (UKCTOCS). Of the 44,799 women who attended annual incidence screening, 960 underwent screen-positive surgery. 80/113 ovarian-tubal cancer were invasive epithelial of these, 37.5% were Stage I/II. 50/52 ovarian-tubal cancer diagnosed within one year of their last screen were invasive epithelial and of these, only 6% were Stage I/II. Data concerning sensitivity, specificity, and positive predictive values of all primary ovarian/tubal cancers or invasive epithelial cancers suggest that transvaginal ultrasound scanning as the first line test might not be suitable for population screening for ovarian cancer.
Authors highlight the major limitation of the study in the subjectivity of ultrasound scanning of postmenopausal ovaries with only moderate agreement for visualisation of normal ovaries in a retrospective random audit. Furthermore the improvement of understanding of ovarian cancer origin has highlighted the importance of evaluation of fallopian tubes that in this study have not been taken into consideration due also to technological limitations.
The manuscripts reports the great effort made to draw conclusion from the incidence screening of the transvaginal ultrasound arm of the UKCTOCS. The main concern of this reviewer is related to the possible presence of subjects with family risk of cancer in the studied population. Are these data known? Surgical samples from patients who underwent surgery have been collected for translational studies? Are there translational studies planned? Biological samples collected from patients undergoing screening strategies can be extremely important also for prognostic studies.
Minor concerns
Table 1 and 2 are difficult to follow, data can be perhaps better shown using a schematic representation.
Supplementary table 1 was not visible.
Author Response
We thank Reviewer 2 for his/her comments and address the questions raised below:
(a) the possible presence of subjects with family risk of cancer in the studied population. Are these data known?
The trial was designed to investigate only those women who were at low or population level risk of developing ovarian cancer. Being at ‘increased risk of familial ovarian cancer’ was specified as one of the exclusion criteria.
Increased risk of familial ovarian cancer was defined as follows: These women are first-degree (1o) relatives (mother, sister, daughter) of a cancer affected member of a “high risk” family.
The high-risk family is defined by one of the following criteria:
- Two or more individuals with OC who were 1o relatives
- One individual with OC and 1 individual with breast cancer diagnosed under
50 years who are 1o relatives
- One individual with OC and 2 individuals with breast cancer diagnosed under
60 years who are connected by 1o relationships
- An affected individual with a mutation of one of the known ovarian cancer
predisposing genes (BRCA1, BRCA2, MSH1)
- One person with OC and three individuals with colorectal cancer with at least
one case diagnosed before 50 years, all connected by 1o relationships.
- Affected relatives fulfilling criteria 1, 2 or 3 who are related by second degree through an unaffected intervening male relative
Volunteers who fitted the above criteria were excluded from the trial.
Of the 50,623 women recruited to the ultrasound arm in UKCTOCS a very small number of women self-reported a maternal history of either breast (3206 (6.3%)) or ovarian cancers. (778 (1.5%))
We have previously published these data (Jacobs, Menon et al (2016) Lancet 387: 945–56.
(b) Surgical samples from patients who underwent surgery have been collected for translational studies? Are there translational studies planned?
The trial protocol did not include banking of surgical samples from women who underwent surgery as a result of a screen positive test. The histopathological tissue blocks are available at the respective NHS Trusts where the women underwent treatment and can be retrieved for translational studies with appropriate ethical approval.
(c) Biological samples collected from patients undergoing screening strategies can be extremely important also for prognostic studies.
We have collated a large collection of serum samples (n = 544,808) during the trial, drawn at baseline from all 202,638 women and then annual samples (median 8) from the women randomised to blood based multimodal screening group.
A number of translational studies have been undertaken. Samples with linked data are available through our data sharing application process.
Our website provides detailed information http://uklwc.mrcctu.ucl.ac.uk/
(d) Table 1 and 2 are difficult to follow, data can be perhaps better shown using a schematic representation.
Most of the data in Table 1 is schematically represented in Figure 2. The difficulty in following the Table may have been due to the loss of formatting. Please see response to Reviewer 1 comments.
- d) Supplementary table 1 was not visible.
We have now uploaded Supplementary Table 1 as a separate document.